# Immunoproteasome Activity and Content Determine Hematopoietic Cell Sensitivity to ONX-0914 and to the Infection of Cells with Lentiviruses

**DOI:** 10.3390/cells10051185

**Published:** 2021-05-12

**Authors:** Elmira Vagapova, Alexander Burov, Daria Spasskaya, Timofey Lebedev, Tatiana Astakhova, Pavel Spirin, Vladimir Prassolov, Vadim Karpov, Alexey Morozov

**Affiliations:** 1Engelhardt Institute of Molecular Biology, Russian Academy of Sciences, Vavilov Street 32, 119991 Moscow, Russia; vr.elmira@gmail.com (E.V.); alexanderburov1998@gmail.com (A.B.); drspssk@gmail.com (D.S.); lebedevtd@gmail.com (T.L.); discipline82@mail.ru (P.S.); prassolov45@mail.ru (V.P.); karpovvl2008@gmail.com (V.K.); 2Koltzov Institute of Developmental Biology, Russian Academy of Sciences, Vavilov Street 26, 119334 Moscow, Russia; tastakhova@bk.ru

**Keywords:** proteasome, immunoproteasome, proteasome inhibitor, ONX-0914, retrovirus, HIV-1, viral infection, cells

## Abstract

Proteasomes are intracellular structures responsible for protein degradation. The 20S proteasome is a core catalytic element of the proteasome assembly. Variations of catalytic subunits generate different forms of 20S proteasomes including immunoproteasomes (iPs), which are present mostly in the immune cells. Certain cells of the immune system are primary targets of retroviruses. It has been shown that several viral proteins directly affect proteasome functionality, while inhibition of proteasome activity with broad specificity proteasome inhibitors stimulates viral transduction. Here we specifically addressed the role of the immunoproteasomes during early stages of viral transduction and investigated the effects of specific immunoproteasome inhibition and activation prior to infection using a panel of cell lines. Inhibition of iPs in hematopoietic cells with immunoproteasome-specific inhibitor ONX-0914 resulted in increased infection by VSV-G pseudotyped lentiviruses. Moreover, a tendency for increased infection of cloned cells with endogenously decreased proteasome activity was revealed. Conversely, activation of iPs by IFN-γ markedly reduced the viral infectivity, which was rescued upon simultaneous immunoproteasome inhibition. Our results indicate that immunoproteasome activity might be determinative for the cellular antiretroviral resistance at least for the cells with high iP content. Finally, therapeutic application of immunoproteasome inhibitors might promote retroviral infection of cells in vivo.

## 1. Introduction

Protein degradation is as important as the protein synthesis. Intracellular hydrolysis of proteins is performed by the ubiquitin-proteasome system (UPS) or by the autophagy depending on the nature, condition, and physical characteristics of substrates [1]. Most short-lived and damaged proteins are degraded by the UPS. The system is tightly regulated, and several levels of its organization exist. The central element of the UPS is the 20S proteasome. Thus, the first level of UPS organization is the composition of the 20S core particle. The 20S proteasome is assembled from pairs of 14 different subunits of two types: the alpha (α1-7) and the beta (β1-7). Three out of seven beta subunits are catalytically active and hydrolyze peptide bonds after acidic (β1), basic (β2), and hydrophobic (β5) amino acids, demonstrating caspase-like, trypsin-like, and chymotrypsin-like activities, correspondingly [2]. In most somatic cells in conditions of stress and inflammation constitutive catalytic subunits are replaced by so-called “immune” subunits β1i, β2i, and β5i during the proteasome assembly. Incorporation of immune subunits leads to the elevation of chymotrypsin-like activity and changes within the pool of generated peptides occur facilitating presentation of antigenic peptides [3,4]. Hence, while being activated in stress conditions in somatic cells, immunoproteasomes (iPs) are constantly expressed in cells of the immune system. Concordantly, Ips participate in the immune reactions and cytokine production. Moreover, immunoproteasomes are involved in the degradation of damaged proteins, regulation of signaling pathways and gene expression, T-cell expansion, retinal function, muscle differentiation, and facilitate maintenance of pluripotency of stem cells [5]. It should be mentioned that there is a subtype of proteasomes known as the intermediate proteasomes. These proteasomes were shown to contain the β5i subunit only or β5i and β1i in combination with constitutive catalytic subunits [6]. Intermediate proteasomes are normally found in different tissues; however, their functions are largely unknown. Nevertheless, intermediate proteasomes were shown to broaden the repertoire of generated peptides enhancing the presentation of specific antigens [6]. Since the β5i subunit is found in both immuno- and intermediate proteasomes, for the sake of simplicity we further referred all β5i-containing proteasomes as immunoproteasomes.

Viruses are supreme parasites capable of using and regulating different cellular systems including the UPS for their own needs [7]. To counteract viral infections cells have developed multiple mechanisms. Proteasomes are actively engaged in these processes. Modulation of proteasome activity using different inhibitors revealed that proteasomes affect viral replication and assembly as well as the efficacy of infection [8,9,10,11,12,13,14]. By hydrolyzing anti-viral host proteins and viral proteins proteasomes may, however, play dual roles during viral infections [15]. When the efficacy of infection is concern, opposite effects of proteasome inhibition were reported for different viruses. Inhibition of proteasome activity reduced Influenza A Virus, Herpes Simplex Virus, and coronavirus infection of permissive cells [10,12,13,16], but stimulated lentiviral infection of lymphocytes [8]. Importantly, in the above-mentioned studies most of the used inhibitors were broad specificity proteasome inhibitors that downregulated the activity of both constitutive and immunoproteasomes, thus did not allow to study the role of particular proteasome forms including the iPs. At the same time, several reports highlight the effect of HIV-1 infection and certain viral proteins on the iP expression and functional state, indicating an important role of this proteasome form in virus–host interactions [17]. Recently subunit-specific proteasome inhibitors became available allowing elucidation of a specific role of iPs in the organism in general and during viral infections in particular. Here we investigated the role of immunoproteasomes during the early stages of retroviral infection using a panel of cell lines, the β5i subunit-specific inhibitor ONX-0914 and the broad specificity proteasome inhibitor–Bortezomib.

## 2. Materials and Methods

### 2.1. Cell Lines

Cell lines U937, HL-60, THP-1, SH-SY5Y, HEK 293, and HEK 293T were cultured in RPMI-1640 (Thermo Fisher Scientific, Paisley, Renfrewshire, Scotland, UK) and DMEM (Thermo Fisher Scientific, Paisley, Renfrewshire, Scotland, UK), respectively, supplemented with 10% fetal bovine serum (FBS) (Hyclone, Logan, UT, USA) and penicillin-streptomycin mixture at 37 °C in a humidified atmosphere containing 5% of CO_2_. The THP-1 cells were presented by Dr. Nikita Nikiforov from the Institute of Experimental Cardiology, National Medical Research Center of Cardiology. HL-60 and U937 cells were a kind gift from Prof. Carol Stocking. SH-SY5Y cells were a generous gift from HPI, Hamburg. HEK 293 and HEK 293T cells were kindly provided by Prof. Boris Fehse.

### 2.2. Cell Viability Assay

The U937, HL-60, THP-1, SH-SY5Y, and HEK 293 cells were treated with 0–200 nM of ONX-0914 and 0–7.5 nM of Bortezomib. Cellular viability was assessed 72 h post-drug-treatment using trypan blue exclusion in Neubauer chamber.

### 2.3. RNA Isolation and Real-Time PCR

The expression levels of genes *PSMB6*, *PSMB7*, *PSMB5*, *PSMB9*, *PSMB10*, *PSMB8*, and *Actb*, encoding proteasome catalytic subunits β1, β2, β5, β1i, β2i, β5i, and β-actin, respectively were quantified using the RT-qPCR system described previously [18]. Total RNA was extracted from THP-1, HL-60, U937, SH-SY5Y, and HEK 293 cells using GeneJet RNA purification kit (Thermo Fisher Scientific, Vilnius, Lithuania), according to the manufacturer’s instructions. The quality and concentration of obtained RNA was assessed using NanoDrop spectrophotometer (Thermo Fisher Scientific, Wilmington, DE, USA). One microgram of total RNA was used to obtain the cDNA using Maxima H Minus Reverse Transcriptase (Thermo Fisher Scientific, Vilnius, Lithuania). Ten nanograms of cDNA were used per reaction. Utilized primers are listed in Appendix A. Gene expression levels were measured in triplicates. qPCR was performed using the Luminaris Color HiGreen qPCR Master Mix kit (Thermo Fisher Scientific, Vilnius, Lithuania) according to the manufacturer’s recommendations. The system calibration was carried out with serial dilutions of the control plasmids (10, 10^2^, 10^3^, 10^4^, 10^5^, and 10^6^ copies of each plasmid was added to the reaction) [18]. Reactions with control plasmids were performed in the presence of 100 ng of carrier DNA (salmon sperm DNA (Thermo Fisher Scientific, Carlsbad, CA, USA)). The qPCR was performed using a LightCycler 480 instrument (Roche, Mannheim, Germany). The output data of the instrument, including the values of the threshold cycles, were analyzed using Microsoft Excel and R tools. Based on the obtained calibration curves (the values of the threshold cycles for serial dilutions of an appropriate plasmid), the number of transcripts of each gene per 10 ng of cDNA was determined. The real-time PCR with primers to *LDLR* and cytokine genes was performed without serial dilutions of control plasmids. The relative gene expression levels were calculated using delta-delta Ct method.

### 2.4. Preparation of Lysates and Western Blotting

Cells were washed three times with PBS, scrapped and homogenized in the NP40 cell lysis buffer (1% NP-40; 150 mM NaCl; 50 mM Tris-HCl pH 8.0). Cellular pellets were mixed with the appropriate volume of buffer and left on ice for 10 min and then centrifuged for 10 min 10,000× *g*. The supernatants were collected and stored at −80 °C before use. The lysates were loaded into the 12% Tris-Glycine PAG and transferred onto the nitrocellulose membrane (Bio-Rad, Hercules, CA, USA). Western blots were performed with either rabbit anti-β1i (Abcam, Cambridge, UK), or rabbit anti-β5i (Abcam, Cambridge, UK) primary antibodies and goat anti-rabbit HRP conjugates (Abcam, UK). Blots were revealed using ECL prime detection kit (GE Healthcare, Little Chalfont, Buckinghamshire, UK), according to the manufacturer’s instructions. For the signal normalization membranes were striped and stained with primary mouse anti-β-actin antibodies (Abcam, Cambridge, UK) and goat anti-mouse HRP conjugates (Enzo, Farmingdale, NY, USA). Blots were revealed as described above. Original Western blot images can be found as Appendix A.

### 2.5. Preparation of Viruses

The VSV-G pseudotyped lentiviral particles were generated by co-transfection of HEK 293T with LeGOC or LeGOG, and packaging plasmids (pMDLg-Addgene plasmid 1225, pRSV-Rev). The LeGO plasmids [19], phCMV-VSV-G [20], pMDLg/pRRE, and pRSV-Rev vectors [21] were a kind gift from Prof. Dr. Boris Fehse and Dr. Christopher Rieken. Condensed medium containing lentiviral particles (LP) was harvested from HEK 293T cells, filtered and stored at −80 °C according to the protocol issued by lentigo-vectors.de. To analyze the quality of viral preps the virus-containing media was concentrated 100 times by centrifugation through the 20% sucrose cushion at 110,000× *g* for 1.5 h. Concentrated samples were analyzed by Western blotting with primary sheep antibodies to HIV-1 p17 and p24 (Dako, Glostrup, Hovedstaden, Denmark) and secondary anti-goat HRP conjugates (Dako, Glostrup, Hovedstaden, Denmark). Blots were revealed as described above. Original Western blot image can be found as Appendix A.

### 2.6. Inhibition of Proteasomes and Evaluation of Viral Infection Efficacy 

Two proteasome inhibitors: broad specificity inhibitor Bortezomib (Selleckchem, Houston, TX, USA), and β5i subunit-specific inhibitor ONX-0914 (Apex bio, Houston, TX, USA) were used in the study. To characterize the effect of proteasome inhibition on the efficacy of viral infection 2 × 10^4^ of cells (THP-1, HL-60, U937, SH-SY5Y and HEK 293) were incubated with 50 nM ONX-0914 or 3 nM Bortezomib for 6 h prior the addition of lentiviral particles. To estimate the viral infection efficacy, we applied previously described system for screening of anti-HIV inhibitors [22]. Briefly, the approach is based on the transfer of reporter gene (encoding fluorescent protein GFP or mCherry) to the host cells with lentiviral particles and the quantification of fluorescence by flow cytometry. The transduction rate is evaluated by comparison of mean fluorescence and per cent of fluorescent cells. To achieve transduction level of 30–50%, 1 to 200 µL of LP-containing medium was added to the cells. The appropriate amounts of virus-containing medium were established for each cell line. Thus, 1 µL was used for HEK 293 cells, 2 µL for the SH-SY5Y cells, 20 µL for the THP-1 cells, 30 µL for the U937 cells, and 200 µL for the HL-60 cells. After that, cells were incubated for an additional 72 h. Total of 2 × 10^5^ cells were collected in 1.5 mL tubes and washed once with 500 µL of PBS. Before the analysis cells were resuspended in 400 µL of PBS. Detection of fluorescence intensity was performed on LSRFortessa flow cytometer (BD Biosciences, San Jose, CA, USA).

### 2.7. Determination of Proteasome Activity

The chymotrypsin-like and beta5i-specific proteasome activities were determined in cellular lysates similarly as described in [23]. In brief, two fluorogenic substrates Suc-LLVY-AMC (Sigma, St. Louis, MO, USA) and Ac-ANW-AMC (Boston Biochem, Cambridge, MA, USA) were used to estimate the chymotrypsin-like and β5i-specific proteasome activities, correspondingly. Aliquots (~6 µL) of lysates were incubated in 100 μL of the reaction buffer (RB), containing 40 mM Tris-HCl (pH 7.5), 1 mM DTT, 5 mM MgCl_2_, 1 mM ATP, and 30 μM of substrate for 20 min at 37 °C. Control reactions with 100 nM of the proteasome inhibitor Bortezomib were performed to test nonspecific degradation of substrates. Reactions were stopped with 2% SDS solution (in ddH2O). Fluorescence at the excitation wavelength 380 nm and emission wavelength 440 nm was measured using VersaFluor Fluorometer (Bio-Rad, Hercules, CA, USA). To calculate the relative activity levels, the activity levels in samples with Bortezomib were subtracted from the values detected in lysates and the obtained values were normalized to one µg of total protein. Proteasome activity in living cells was determined using cell-permeable proteasome activity probe Me4BodipyFL-Ahx3Leu3VS (UbiQbio, Amsterdam, The Netherlands) according to the protocol described in [24].

### 2.8. Treatment of Cells with IFN-γ

The HL-60, THP-1, U937, SH-SY5Y, and HEK 293 cells were treated with 1000 U/mL of recombinant human IFN-γ (R&D Systems, Minneapolis, MN, USA) for 48 h and further either analyzed for the proteasome subunit mRNA and protein expression, proteasome activity or transduced with lentiviruses. When transduction efficacy of cells pre-treated with proteasome inhibitors and stimulated with IFN-γ was studied, THP-1 cells were first treated with 1000 U/mL of IFN-γ for 42 h then either 50 nM of ONX-0914 or 3 nM of Bortezomib was added into the cell culture media and after additional 6-h incubation viruses were introduced. Cells were incubated for an additional 72 h and after that viral infection efficacy was estimated as described above.

### 2.9. Statistical Analysis and Software

Bar carts depicts mean values ± standard deviation for experimental replicates. The two-way ANOVA test for multiple comparisons with no correction was used to compare three and more groups. Where indicated the unpaired two-tailed *t*-test was used to evaluate the statistical significance of differences between the experimental groups. The obtained flow cytometry data were analyzed using FlowJo version 10.0.7 (FlowJo LLC, Becton Dickinson and Company, Franklin Lakes NJ, USA) and GraphPad Prism 8. 4.3. (GraphPad Software, San Diego, CA, USA) software. ImageJ (US National Institutes of Health, Bethesda, MD, USA) software was used for the quantification of blots.

## 3. Results

### 3.1. Different Cell Lines Demonstrate Different Viability Following Treatment with Proteasome Inhibitors

To investigate the effect of proteasome inhibitors on viral infection of different cell lines we initially studied the viability of hematopoietic (HL-60, THP-1, U937) neuronal (SH-SY5Y) and kidney (HEK 293) cells after incubation with different concentrations of β5i-specific inhibitor ONX-0914 and broad specificity proteasome inhibitor Bortezomib. Comparing with the SH-SY5Y and HEK 293 cells, hematopoietic cell lines were considerably more sensitive to ONX-0914 (Figure 1a). Among the latter, monocytic THP-1 cells demonstrated highest sensitivity to ONX-0914 with the IC50 of 47.7 nM (Figure 1a). At the same time, SH-SY5Y and HEK 293 cells did not show decreased viability in response to 150 nM of this inhibitor (Figure 1a). When cell lines were incubated with Bortezomib, SH-SY5Y cells were found to be the most sensitive with the IC50 of around 2.8 nM (Figure 1b). The IC50 values for other cells after Bortezomib treatment ranged from 4.5 to 5 nM except for HEK 293 cells, which were found insensitive to 7.5 nM of the inhibitor (Figure 1b).

Based on the obtained data, 50 nM and 3 nM concentrations of ONX-0914 and Bortezomib, respectively, were selected for the further experiments.

### 3.2. Proteasome Subunit Expression and Activity in THP-1, HL-60, U937, SH-SY5Y, and HEK 293 Cell Lines

To associate the effects of proteasome inhibitors on viability with intracellular proteasome pool diversity we examined the expression of catalytic proteasome subunits and evaluated the overall chymotrypsin-like, as well as the β5i-specific activities in the cells. It was previously shown that among the tested cell lines, U937 contains mostly proteasomes with the immune subunits, while constitutive proteasomes dominate in the pool of HEK 293 cells [25]. Thus, we first determined the number of transcripts of genes encoding catalytic proteasome subunits (constitutive *PSMB5*, *PSMB6*, *PSMB7* and immune *PSMB8*, *PSMB9*, *PSMB10*) within the tested cells (Figure 2a,b). Concordantly with [25] constitutive proteasome subunits transcripts were abundantly revealed in HEK 293 cells with the highest expression levels detected for the *PSMB5* gene (encodes β5 subunit). Comparing to SH-SY5Y or HEK 293 cells, significantly higher levels of *PSMB8* (encodes β5i subunit) (*p* < 0.0001, ANOVA) and *PSMB9* (encodes β1i subunit) (*p* < 0.01, *p* < 0.001, *p* < 0.0001, ANOVA) mRNA were revealed in hematopoietic cells (Figure 2b). Although immunoproteasome gene expression might not directly correlate with the amount of specific protein [26], these results favor high amounts of proteasomes with β5i and low number of constitutive proteasomes in hematopoietic cells and, conversely, prevailing amounts of proteasomes with constitutive beta subunits in HEK 293 and SH-SY5Y cells.

The β5, β5i, and β1i subunits are responsible for the chymotrypsin-like activity of proteasomes. Along these lines, we have compared general chymotrypsin-like activity and β5i-specific activity in tested cells using Suc-LLVY-AMC and Ac-ANW-AMC substrates for chymotrypsin-like and β5i-specific proteasome activities, respectively. The highest chymotrypsin-like activity was revealed in THP-1 cells (*p* < 0.0001, ANOVA), while in other cell lines the activity was comparable (Figure 3a). In hematopoietic cells the β5i-specific activity was from 2.5 to 20 times higher than in SH-SY5Y or HEK 293 cells, the highest being in THP-1 cells (*p* < 0.0001, ANOVA) (Figure 3b). These results demonstrate that the proportion of immunoproteasome activity in overall chymotrypsin-like activity is significantly higher within the hematopoietic cells than within the SH-SY5Y or HEK 293 cells.

Next, we have characterized the changes of proteasome activities induced by 50 nM of ONX-0914 and 3 nM of Bortezomib following 6 h of incubation (Figure 3a,b). In THP-1 cells incubated with ONX-0914 the chymotrypsin-like and β5i-specific activities were decreased by 90% and 94%, correspondingly (*p* < 0.0001, ANOVA). In HL-60 and U937 cells treated with ONX-0914 the chymotrypsin-like activity was decreased by 70% and 80%, while the β5i-specific activity—by 83% and 86%, respectively (*p* < 0.0001, ANOVA). Used concentration of Bortezomib also induced a decrease of chymotrypsin-like and β5i-specific activities in hematopoietic cells, but to a lower extent than ONX-0914. The ONX-0914 had no effect on the chymotrypsin-like activity in SH-SY5Y or HEK 293 cells, while Bortezomib significantly decreased the chymotrypsin-like proteasome activity in SH-SY5Y (*p* < 0.0001, ANOVA), but not in HEK 293 (Figure 3a,b). Obtained data indicate that the β5i-containing proteasomes are responsible for most of the chymotrypsin-like activity in hematopoietic cells and that these proteasomes are efficiently inhibited by the chosen concentration of ONX-0914 (Figure 3a,b). At the same time, the proportion of active immunoproteasomes in neurons and HEK 293 cells is negligible and inhibition of these proteasomes by ONX-0914 minimally affects overall cellular chymotrypsin-like activity. These results match the toxicity data, seemingly explain the sensitivity of the hematopoietic cells, especially of THP-1 cells to the ONX-0914 and are in congruence with the reported data [27].

### 3.3. Cells with High Immunoproteasome Activity Demonstrate Enhanced Infection with Lentiviruses after Treatment with ONX-0914 

To study the transduction efficacy of cells viral preparations were obtained and characterized by Western blotting. The presence of cleaved HIV-1 Gag proteins p17 and p24 in 100× concentrated virus-containing media was demonstrated (Figure 4a). To evaluate if ONX-0914 or Bortezomib treatment alters the infection efficacy of cells, THP-1, HL-60, U937, SH-SY5Y, and HEK 293 cells were incubated with proteasome inhibitors for 6 h. After that, appropriate amounts of virus-containing media, sufficient to achieve transduction level of 30–50% of a particular cell line were introduced and after additional 72 h incubation the transduction efficacy was measured (Figure 4b).

Bortezomib moderately stimulated transduction of THP-1 cells (*p* < 0.0001, ANOVA). On the contrary, a significant increase of transduction levels was revealed in THP-1 (*p* < 0.0001, ANOVA), HL-60 (*p* < 0.001, ANOVA), and U937 (*p* < 0.01, ANOVA) cells incubated with ONX-0914 (Figure 4b). Thus, the transduction efficacy of THP-1 and HL-60 cells was increased from 49% to 76% and from 36% to 56%, correspondingly. The infection efficacy of U937 cells was increased by approximately 10 per cent following treatment with the ONX-0914 (Figure 3b). At the same time, viral transduction levels did not change after the treatment of HEK 293 cells or changed marginally following the incubation of SH-SY5Y cells with the inhibitors.

Recently several proteasome activity-based probes were developed allowing the estimation of proteasome activity in lysates, but also in living cells [24,28]. To make sure that increased transduction is happening against the background of constantly reduced proteasome activity, 72 h after the treatment with ONX-0914 and Bortezomib the proteasome activity was measured in THP-1, HL-60, and U937 cells simultaneously with the infection efficacy using fluorescence-based cell-permeable proteasome activity probe. It was shown that proteasome activity remained decreased in the ONX-0914-treated THP-1 cells (*p* < 0.0001, ANOVA) at the time of viral infection efficacy assessment (Figure 4c).

It should be mentioned that since both ONX-0914 and Bortezomib affect immunoproteasomes, in cells with predominant immunoproteasome content at certain concentrations comparable effects may be expected. However, the normalization of the inhibitors concentrations is challenging due to several reasons including toxicity, specificity, and targeting of constitutive proteasomes by Bortezomib. Nevertheless, by using lower concentrations of ONX-0914 and higher concentrations of Bortezomib we managed to obtain comparable transduction levels of THP-1 cells (Appendix A), however the degree of proteasome inhibition (Appendix A) and condition of cells were significantly different.

The obtained results indicate that the efficacy of viral infection might be dependent on the cellular proteasome activity and on the presence of proteasomes with immune subunits within the intracellular pool. Moreover, in hematopoietic cells viral transduction efficacy increase might be associated with the reduction of the immunoproteasome activity.

However, performed experiments do not allow exclusion of potential side effects of inhibitors on the transduction efficacy. In order to minimize this possibility, we obtained six clones derived from the U937 cell line. It was shown that clones have the same division rate (data not presented). Obtained clones were transduced with viruses and varying transduction efficacies were revealed (Appendix A). The lowest transduction efficacy was shown for the F11-1 and G12-5 clones, the average for H9-6, C8-6, and F12-6, the highest (app. 160% to the level in F11-1) was observed for the D9-6 clone (Appendix A). Next, we estimated the chymotrypsin-like activity in the clones and calculated the correlation between the activity and viral transduction efficacy (Appendix A). Although the correlation was not significant a clear trend indicating that increased viral transduction is associated with decreased chymotrypsin-like activity was observed (Appendix A). Indeed, the F11-1 clone with the highest chymotrypsin-like proteasome activity (*p* < 0.001) (Appendix A), demonstrated the lowest transduction rate (*p* < 0.05) among cell clones (Appendix A). Moreover, expression of proteasome subunit genes *PSMB8* and *PSMB5* was higher in C8-6 clone, which demonstrated lower transduction efficacy than in the D9-6 clone with the highest transduction level (Appendix A). Since differences in the transduction efficacy of cells by the VSV-G pseudotyped lentiviral particles might be associated with the altered amount of low-density lipoprotein receptors (LDLR) on the cellular surface [29], we evaluated the expression of the corresponding gene within the clones. The LDLR mRNA expression levels in F11-1, D9-6, and C8-6 clones were comparable, indicating that different transduction efficacies of clones do not correlate with the receptor gene expression levels (Appendix A). Together, obtained results favor association of proteasome activity levels with the transduction efficiency.

### 3.4. IFN-γ Increases Proteasome Activity and Affects Viral Infection

To investigate if the induction of proteasome expression can abrogate viral infection, we used HEK 293 cells with low iP content and induced overexpression of immunoproteasome subunits within the cells. The pcDNA3.1- was used as a backbone and expression vectors encoding iP subunits β1i, β2i, and β5i were obtained. HEK 293 cells were transfected with the vectors and 48 h post transfection the expression of immunoproteasome subunits was estimated by Western blotting (Appendix A). Efficient expression of recombinant proteasome subunits was demonstrated. However, very low amount of cleaved iP subunits was revealed. Since at the end of 20S proteasome assembly catalytic subunits undergo auto-catalytic cleavage of propeptides, integrated proteasome subunits have lower molecular weight than the precursor molecules. In lysates of transfected cells little of cleaved immune subunits was detected, indicating efficient expression, but inefficient integration of proteasome subunits (Appendix A). HEK 293 cells were co-transfected with the obtained plasmids and after 48 h of incubation virus-containing media was added. We detected no significant difference in the transduction efficacy between transfected and non-transfected cells (data not shown). The absence of effect might be associated with inefficient integration of the synthesized subunits into the immunoproteasomes (Appendix A), therefore we sought to use another approach to promote iP synthesis.

It is well established that recombinant IFN-γ stimulates expression of immunoproteasome genes in cultured cells [30]. Thus, HL-60, THP-1, U937, SH-SY5Y, and HEK 293 cells were treated with 1000 U/mL of recombinant human IFN-γ for 48 h. The expression of proteasome subunits mRNA and corresponding protein levels were characterized (Figure 5a,d). qRT-PCR analysis demonstrated elevation of *PSMB8*, *PSMB9*, *PSMB10*, but not *PSMB5* mRNA levels in HL-60, THP-1, U937, HEK 293, and SH-SY5Y cells (Figure 5a). Interestingly, the upregulation of *PSMB9* expression was the highest, while the induction of *PSMB10* (encodes β2i), the weakest.

Along these lines Western-blot analysis of cell lysates confirmed increased levels of integrated β1i (encoded by the *PSMB9*) in all the examined cells following the IFN-γ exposure (Figure 5b,c). The β1i expression was most efficiently stimulated in THP-1, HL-60, SH-SY5Y, and HEK 293 cells. Here it should be mentioned that despite considerable upregulation, β1i levels in HEK 293 cells were very low. The IFN-γ-stimulated induction of β5i (encoded by the *PSMB8*) expression in U937 cells was limited while it was considerable in SH-SY5Y, HL-60, and THP-1 cells. Nevertheless, β5i levels in SH-SY5Y remained quite low after the treatment with IFN-γ and no upregulation of β5i in HEK 293 cells was revealed (Figure 5b,c). Concordantly, general chymotrypsin-like proteasome activity and β5i-specific activity were significantly increased in THP-1 cells being insignificantly affected in HEK 293 cells (Figure 5d). In a parallel experiment 48 h after IFN-γ treatment virus-containing media was added to the cells and after additional 72 h of incubation the transduction efficacy was assessed. Decreased infection of all the IFN-γ-treated cells was observed (Figure 5e). However, the degree of infection attenuation was different between the cell lines. The effect of IFN-γ on the infection of U937 cells was within 10% and statistically insignificant. The attenuation of HEK 293 and HL-60 (*p* < 0.1, *t*-test) cells infection was roughly 23–25%, however, the transduction of THP-1 cells was almost completely abrogated (*p* < 0.0001, *t*-test) (Figure 5e). Using cell-permeable proteasome activity probe it was demonstrated that 72 h post infection and 120 h post administration of IFN-γ the proteasome activity was increased in THP-1 (*p* < 0.0001, *t*-test) and HL-60 but was almost unchanged in HEK 293 cells (Figure 5e). To confirm the role of iPs in IFN-γ-induced effects we repeated the transduction experiment using cells treated with both the IFN-γ and ONX-0914. It has been shown that in THP-1 cells treated with both ONX-0914 and IFN-γ the transduction efficacy was increased by 70% while in cells treated with the ONX-0914 only, the transduction level was increased by 48% (*p* < 0.0001, ANOVA) (Figure 5f). These results confirm that decrease of transduction efficacy induced by IFN-γ is at least in part dependent on the immunoproteasome activity modulation.

### 3.5. Proteasome Inhibitors Modulate Cytokine Expression in Studied Cell Lines

It was demonstrated that inhibition of proteasomes with broad specificity inhibitors stimulates the expression of pro-inflammatory cytokines in different cells [31,32,33]. Thus, we measured the expression of *IL-1β*, *IL-6*, *CXCL8*, and *TNF-α* in cells treated with ONX-0914 and Bortezomib (Figure 6). In hematopoietic cells ONX-0914 and to a lesser extent Bortezomib influenced cytokine expression. Bortezomib stimulated *IL-1β*, *IL-6*, and *TNF-α* expression in U937 cells and decreased *TNF-α* expression in SH-SY5Y, THP-1, and HEK 293 cells, as well as *IL-1β* in THP-1 and SH-SY5Y cells. At the same time, treatment with ONX-0914 increased the *IL-1β* and *TNF-α* mRNAs content 2–3 folds in THP-1 cells (*p* < 0.0001, ANOVA). In HL-60 cells, however, ONX-0914 induced no changes of *IL-1β* expression but stimulated two-fold *TNF-α* mRNA synthesis (*p* < 0.0001, ANOVA). However, the most prominent stimulation after ONX-0914 treatment was observed for the *CXCL8* mRNA levels. Thus, while in SH-SY5Y the increase of *CXCL8* mRNA content was two-fold, in HL-60—four-fold (*p* < 0.001, ANOVA), in U937—six-folds (*p* < 0.0001, ANOVA), it was 130-folds in THP-1 cells (*p* < 0.0001, ANOVA) (Figure 6). Used concentrations of Bortezomib did not influence the *CXCL8* expression indicating a possible role of targeted iP inhibition for the induction of the cytokine expression. Interestingly, decreased *IL-1β* and *TNF-α* mRNA levels were revealed after ONX-0914 treatment in non-hematopoietic cells (Figure 6).

## 4. Discussion

Retroviruses are unique due to utilization of reverse transcriptase and integrase to deliver their genetic material into the cellular genome in order to propagate. These viruses are so effective that approximately 8% of human genome is composed of retroviruses or their fragments [34]. In millions of years of coevolution viruses have developed several mechanisms to overthrow cellular defense systems. HIV-1 is a global threat and the most studied virus: in 2017 there were 232,000 documents on HIV/AIDS being already published [35]. Thus, it attracted much attention with regards to existing restriction factors and escape mechanisms. Currently there are several cellular proteins known as restriction factors that have been extensively studied with regards to HIV-1 infection: TRIM5α; TRIM33; SAMHD1; MARCH8, APOBEC3 proteins and tetherin [36,37,38]. Interestingly, TRIM5α, TRIM33, and MARCH8 are ubiquitin ligases, while the proper activity of other restriction factors is dependent on the correct function of the UPS which is frequently hijacked by the viral proteins [39]. Thus, the HIV-1 Vif protein was shown to induce ubiquitination and degradation of APOBEC3G [40]. Vpu was demonstrated to interact with tetherin, stimulating its ubiquitination and degradation [41]. HIV-2 protein Vpx has been shown to stimulate ubiquitination and degradation of SAMHD1 [42]. Moreover, HIV-1 proteins are known to modulate proteasome activity. The Tat protein was shown to inhibit the 20S proteasome activity and its association with the 11S regulator [43]. Interestingly, Tat was demonstrated to bind both constitutive and immunoproteasomes through different subunits including β2i, and β5i [44]. In dendritic cells HIV-1 p24 has been shown to downregulate PA28, and iP subunits β2i, and β5i [45]. In human monocyte-derived macrophages the virus was demonstrated to reduce proteasome activity and immunoproteasome expression [46]. Together, accumulating evidence indicate the important role of iPs in virus-host interactions. At the same time, the role of the immunoproteasomes remained undefined till recently when first insights into the specific role of the iPs in HIV-1 infection and restriction were reported. In an interesting study Jimenez-Guardeño et al. revealed that IFN-α-induced upregulation of iPs stimulates turnover of TRIM5α and confers HIV-1 sensitive to the TRIM5α-mediated restriction [47].

In a present study we have demonstrated that immunoproteasome activity might be considered as an additional restriction factor active at least during the early steps of lentiviral infection. We have used five cell lines of different origin and revealed most significant effects of iP inhibition on the lentiviral transduction of hematopoietic cells where the immunoproteasome expression is high. Although we cannot exclude that in cells where there are low numbers of iPs constitutive proteasomes counteract viral infection, our data give insights into the specific function of the immunoproteasomes in hematopoietic cells. Our results are in good congruence with the reported data [47] and previous findings by Schwartz et al., [8] and indicate that immunoproteasome activity is essential for the prevention of hematopoietic cell infection with lentiviruses. Indeed, Schwartz and coauthors initially demonstrated that treatment of different cells with proteasome inhibitors resulted in the increased efficiency of HIV-1 infection [8]. Moreover, it has been shown that following proteasome inhibition HIV-1 Gag proteins accumulated in the cytoplasm as well as the reverse transcription products [8]. The latter fact was confirmed in further studies and associated with the TRIM5α-mediated restriction [11,48]. At the same time in conditions of proteasome inhibition, accumulation of viral cDNA did not lead to the increased proviral integration, likely due to the attenuated disintegration of viral capsids. These results, however, were obtained using broad specificity proteasome inhibitors (target both constitutive and immunoproteasomes) and did not allow elucidation of the specific role of the immunoproteasomes in these processes. To resolve this issue, we have used the ONX-0914—a relatively new β5i-specific proteasome inhibitor [49]. The ONX-0914 is expected to be used for the treatment of autoimmune diseases and its analog is evaluated in phase II clinical trials (https://www.clinicaltrials.gov/ct2/show/NCT03393013 accessed on 15 April 2021). For the experiments we have selected 50 nM concentration of ONX-0914 which was minimally toxic to the used cells and should not affect the activity of constitutive proteasomes since according to [49] concentrations of ONX-0914 less than 200 nM have negligible impact on the constitutive proteasomes. To link our results with the data published by Schwartz et al., [8] we have also included broad specificity proteasome inhibitor Bortezomib in our studies. As expected Bortezomib increased viral infectivity in hematopoietic cells, however chosen concentration of ONX-0914 stimulated viral transduction more efficiently (Figure 4). This coincided with the increased (in comparison with HEK 293 and SH-SY5Y cells where the effect of ONX-0914 was minimal) expression of immunoproteasome subunits and activity in THP-1, HL-60, and U937 cells (Figure 2 and Figure 3). To gain additional confirmation that the proteasome (immunoproteasome) activity is essential for the viral infection we obtained clones of the U937 cell line with different basal levels of proteasome activity. Proteasome pool of the U937 cells was characterized in detail by Fabre et al. and it was demonstrated that β5i-containing proteasomes represent the main proteasome form in these cells [25]. Accordingly, we obtained evidence in favor of higher transduction efficiency of U937 clones with decreased proteasome activity (Appendix A). In addition, we sought to confirm iP-dependent restriction of HIV-1 through the stimulation of proteasome expression. Overexpression of immunoproteasome subunits in HEK 293 cells that practically lack immunoproteasomes [25] had no effect on the viral transduction. This might be due to the inefficient integration of synthesized immune subunits into the assembling proteasomes (Appendix A). Thus, we utilized another approach and stimulated iP expression via incubation of cells with IFN-γ [30]. We have shown that IFN-γ almost completely abrogated the lentiviral infection of THP-1 cells and significantly reduced the transduction level of HL-60 cells, which was accompanied by the increased immunoproteasome subunit expression and activity (Figure 5). At the same time, antiviral mechanisms of IFN-γ seem to require proteasome activity but have not been fully characterized [50]. Concordantly iP-independent pathways cannot be excluded; indeed, IFN-γ decreased the viral infection of HEK 293 cells where immunoproteasome stimulation by this cytokine was marginal. However, the degree of infection attenuation in HEK 293 cells was incomparable with that in THP-1 cells (Figure 5). Moreover, we have shown that in THP-1 cells treated with IFN-γ and ONX-0914 viral transduction is increased more efficiently than in cells treated with the ONX-0914 only (Figure 5f). Thus, our data are in good congruence and expend the results obtained with IFN-α which was also shown to stimulate iP expression and induce TRIM5α-dependent restriction of HIV-1 infection [47]. Since both interferons activate immunoproteasomes and TRIM5α is expressed in most of the studied cell lines (https://www.proteinatlas.org/ENSG00000132256-TRIM5/cell#rna accessed on 15 April 2021) identical mechanisms might be anticipated. At the same time, Jimenez-Guardeño et al. did not detect the effect of ONX-0914 on the infection of cells that were not stimulated with IFN-α, however, the concentration of ONX-0914 that was used is markedly different than that in our experiments. Interestingly, inhibition of proteasomes may influence the degradation of the viral integrase induced by the ubiquitin-ligase TRIM33 [37]. Thus, in condition of proteasome inhibition accumulated viral cDNA molecules might have better chances to integrate even though the viral capsid is disintegrated less efficiently. Moreover, proteasome-dependent degradation of integrase might be the reason behind reciprocal effects of proteasome inhibition on the early steps of lentiviral infection and infection by other RNA and DNA viruses. Since our experiments were focused on the early infection steps, obtained results complement previous findings and do not exclude that proteasome activity is necessary for the efficient Gag polyprotein processing, release, and maturation of HIV-1 virions [9,51].

Since modulation of proteasome activity might affect the inflammatory state of cells, we have investigated the effects of ONX-0914 and Bortezomib on the cytokine production. Both inhibitors were mostly shown to stimulate cytokine genes expression in hematopoietic cells and decrease the cytokine mRNA levels in non-hematopoietic cells (Figure 6). Interestingly, in THP-1 and U937 cells opposite effects of ONX-014 and Bortezomib were demonstrated. In THP-1 cells ONX-914 increased IL1-β and TNF-α mRNA levels, while Bortezomib decreased the expression of the cytokines. On the contrary, Bortezomib but not ONX-0914 stimulated IL1-β and TNF-α expression in U937 cells. At the same time, the degree of stimulation and the spectrum of affected cell lines were larger for the ONX-0914. Altogether, the most significant effects of ONX-0914 were revealed in THP-1 cells. Thus, under the influence of ONX-0914 the *CXCL8* expression was increased by 130-folds (Figure 6). These results demonstrate possible implication of different proteasome forms in regulation of different cytokines and highlight the role of immunoproteasomes in modulation of cytokine expression. Our findings are in good congruence with data obtained previously using broad specificity proteasome inhibitors on A549 cells [31], human hepatocytes [32], prostate cancer cells [33], triple negative breast cancer cells [52], ovarian cancer cells [53], and lung epithelial and monocytic cells including THP-1 [54]. At the same time, decreased *CXCL8* expression following ONX-0914 treatment was reported in human myometrial cells and tissues [55]. Though such discrepancy might be associated with differences in cellular origin and microenvironment. 

## 5. Conclusions

Taken together our data highlight the novel functional activities of immunoproteasomes as a restriction factor for the lentiviral infection and important regulatory pathway defining the cytokine expression in hematopoietic cells. In this regard, we speculate that application of immunoproteasome inhibitors in vivo might create favorable conditions for the lentiviral infection of hematopoietic cells.

## Figures and Tables

**Figure 1 cells-10-01185-f001:**
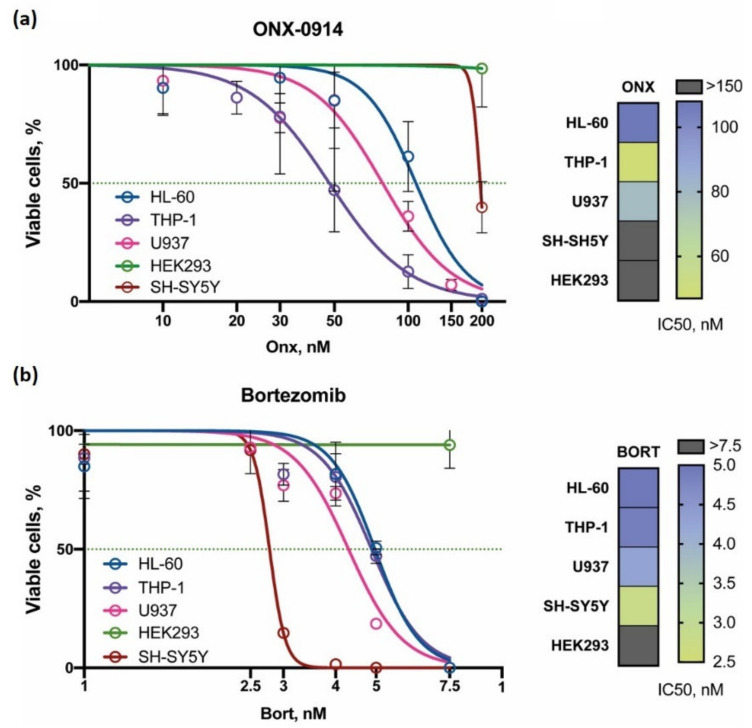
Viability of cells treated with ONX-0914 or Bortezomib. U937, HL-60, THP-1, SH-SY5Y, and HEK 293 were treated with 0–200 nM of ONX-0914 (**a**) and 0–7.5 nM of Bortezomib (**b**). Cellular viability was assessed 72 h post drug-treatment using trypan-blue exclusion. THP-1 cells were the most sensitive to ONX-0914, while SH-SY5Y cells demonstrated highest sensitivity to Bortezomib. Data are represented as average of SEM of five experiments.

**Figure 2 cells-10-01185-f002:**
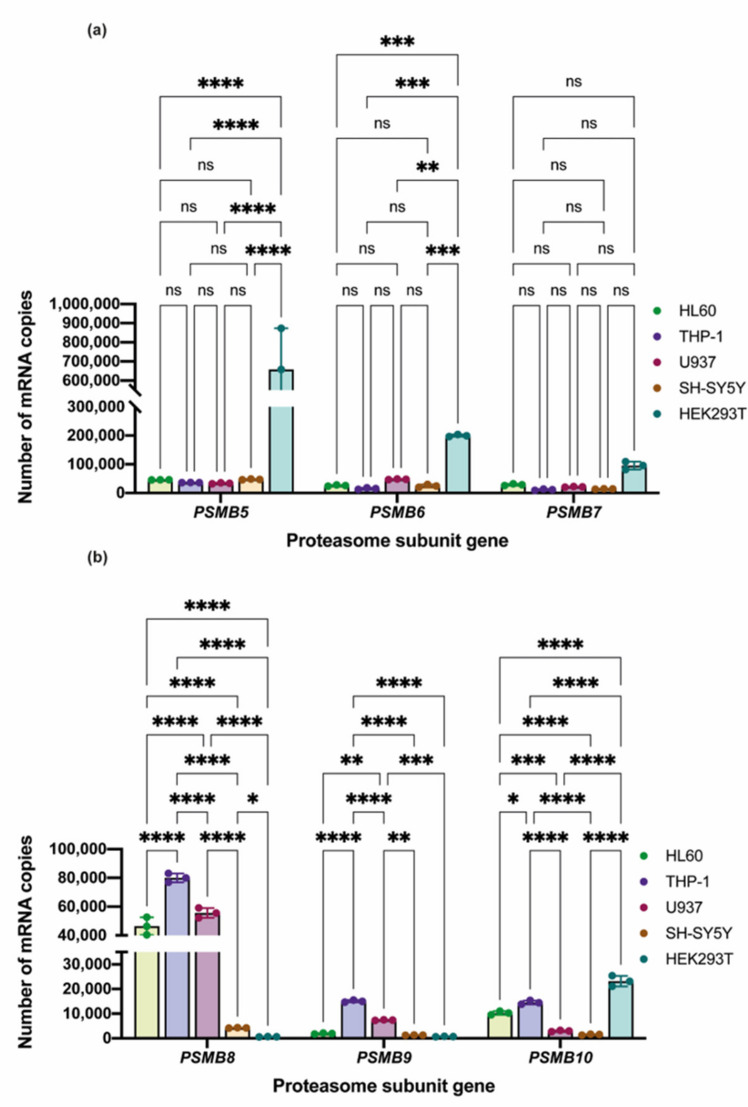
The expression of proteasome genes in studied cell lines. Comparative analysis of constitutive (*PSMB5*, *PSMB6*, and *PSMB7*) (**a**) and immune (*PSMB8*, *PSMB9*, and *PSMB10*) (**b**) proteasome subunits genes expression levels in studied cell lines. The number of copies is specified for 10 ng of cDNA derived from isolated total RNA. Three technical repeats were performed for each sample. Dots represent individual replicates. Two-way ANOVA test for multiple comparisons with no correction was used. Asterisks: * *p*-value < 0.1; ** *p*-value < 0.01; *** *p*-value < 0.001; **** *p*-value < 0.0001; ns—not significant.

**Figure 3 cells-10-01185-f003:**
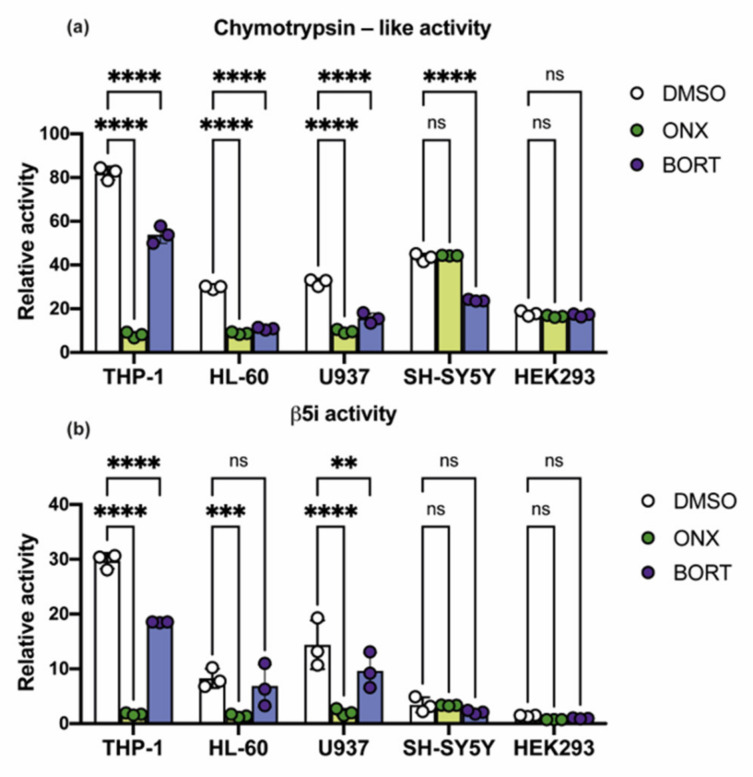
Chymotrypsin-like and β5i-specific activities in the studied cell lines and effects of proteasome inhibitors. Comparative analysis of chymotrypsin-like (**a**) and β5i-specific (**b**) activities in the studied cell lines. Proteasome activities were determined in cellular lysates using two fluorogenic substrates Suc-LLVY-AMC for the chymotrypsin-like activity and Ac-ANW-AMC for the β5i-specific activity after the 6 h incubation with 50 nM of ONX-0914 or 3 nM of Bortezomib. Proteasome activities are shown as relative values, normalized to one µg of total protein. The decrease of chymotrypsin-like activity was revealed in all the studied cell lines except HEK 293 after the incubation with Bortezomib. After treatment with ONX-0914 the chymotrypsin-like proteasome activity was significantly decreased in hematopoietic cells but not in SH-SY5Y or HEK 293 cells. The β5i-specific activity was affected by ONX-0914 in hematopoietic cells. Dots represent the individual replicates. Bars represent standard deviation. Tests were performed in triplicates. Two-way ANOVA test for multiple comparisons with no correction was used. Asterisks: ** *p*-value < 0.01; *** *p*-value < 0.001; **** *p*-value < 0.0001; ns—not significant.

**Figure 4 cells-10-01185-f004:**
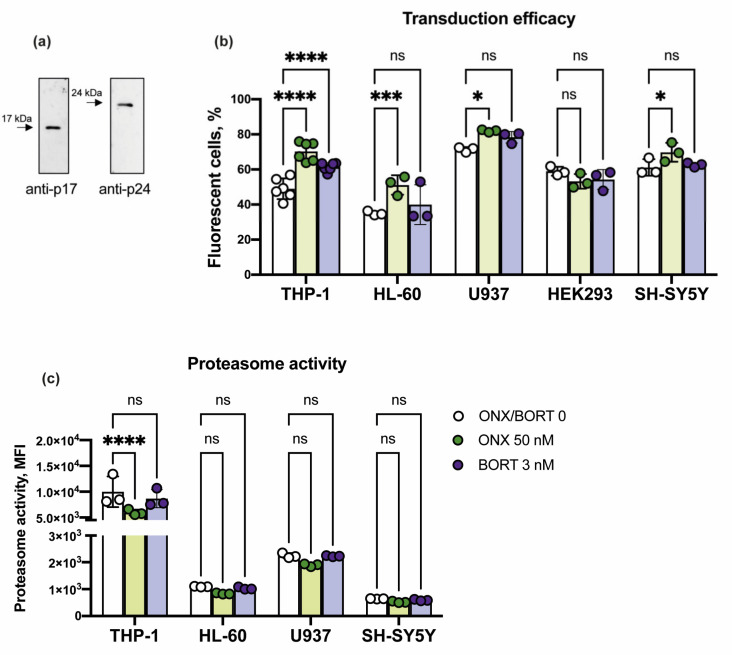
Lentiviral transduction of cell lines following treatment with ONX-0914 or Bortezomib. (**a**) Western blotting of 100 times concentrated virus-containing media with antibodies to HIV-1 p17 and p24. (**b**) THP-1, HL-60, U937, SH-SY5Y, and HEK293 cells were incubated with 50 nM of ONX-0914 or 3 nM of Bortezomib for 6 h prior to the addition of lentiviral particles. After the administration of viruses, cells were incubated for additional 72 h. Cells were collected and fluorescence intensity was measured using LSRFortessa flow cytometer (BD Biosciences). The THP-1, HL-60, and U937 cells demonstrated increased level of transduction following incubation with the ONX-0914. (**c**) Proteasome activity was measured using cell-permeable proteasome activity probe at the time of transduction efficacy estimation. Dots represent individual replicates. Bars represent standard deviation. At least three independent repeats were performed. Two-way ANOVA test for multiple comparisons with no correction was used. Asterisks: * *p*-value < 0.1; *** *p*-value < 0.001; **** *p*-value < 0.0001; ns—not significant.

**Figure 5 cells-10-01185-f005:**
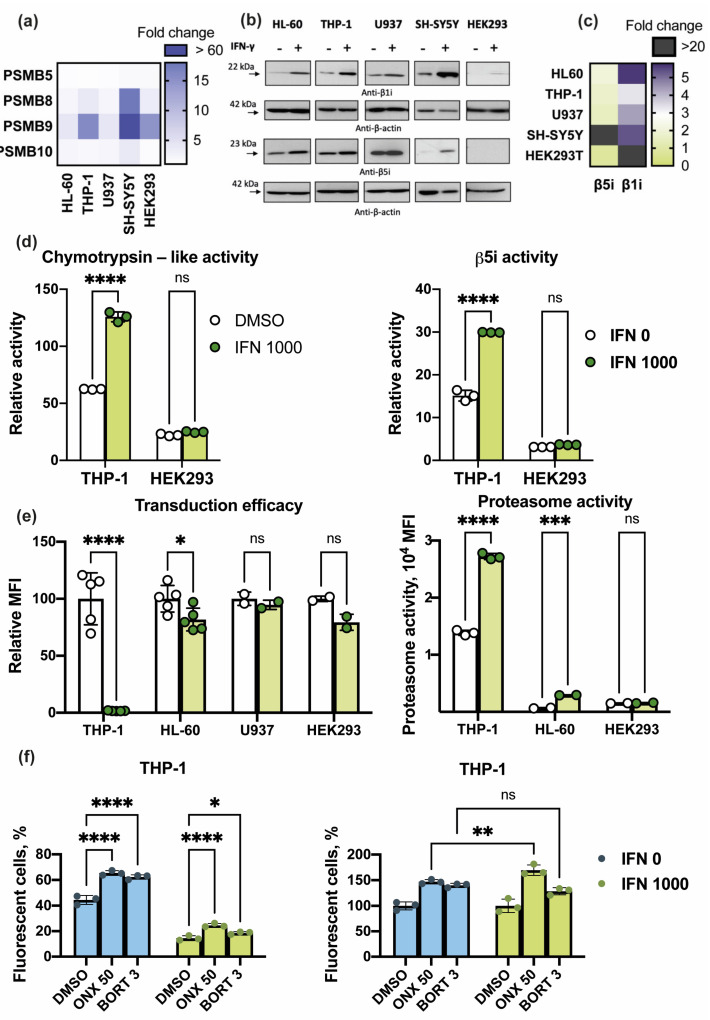
The effects of IFN-γ on the immunoproteasome subunit expression and activity in the studied cell lines and on the efficacy of lentiviral transduction. The HL-60, THP-1, U937, SH-SY5Y, and HEK 293 cells were treated with 1000 U/mL of recombinant human IFN-γ for 48 h and further either analyzed for the proteasome subunit mRNA and protein expression, proteasome activity or transduced with lentiviruses. (**a**) Changes in *PSMB5*, *PSMB8*, *PSMB9*, *PSMB10* mRNA expression levels within HL-60, THP-1, HEK 293, and SH-SY5Y cells after treatment with 1000 U/mL of IFN-γ. Color indicates fold change relative to non-treated cells. (**b**) Western blotting of cellular lysates with antibodies to β1i and β5i. The lysates were obtained from cells treated with 1000 U/mL of IFN-γ and control cells. Corresponding diagram represents quantification of obtained data with the ImageJ software (**c**). (**d**) Chymotrypsin-like (left panel) and β5i-specific activity (right panel) in THP-1 and HEK 293 cells 48 h after the incubation with 1000 U/mL of IFN-γ. Proteasome activities were determined in cellular lysates using two fluorogenic substrates Suc-LLVY-AMC for the chymotrypsin-like activity and Ac-ANW-AMC for the β5i-specific activity. (**e**) Transduction efficacy of cells pre-treated with 1000 U/mL of IFN-γ for 48 h (relative MFI is shown) (left panel); proteasome activity measured using cell-permeable proteasome activity probe at the time of transduction efficacy estimation (right panel). (**d**,**e**) Bars represent standard deviation. Tests were performed in triplicates. Unpaired *t*-test was used for evaluation of significance. Asterisks: * *p*-value < 0.1; *** *p*-value < 0.001; **** *p*-value < 0.0001; ns—not significant. (**f**) Transduction efficacy of THP-1 cells pre-treated with proteasome inhibitors (either 50 nM of ONX-0914 or 3 nM of Bortezomib) and either stimulated or not stimulated with the 1000 U/mL of IFN-γ for 48 h (left panel). The same data normalized to the level of transduction in DMSO-treated cells (right panel). Dots represent individual replicates. Bars represent standard deviation. Three independent repeats were performed. Two-way ANOVA test for multiple comparisons with no correction was used. Asterisks: * *p*-value < 0.1; ** *p*-value < 0.01; *** *p*-value < 0.001; **** *p*-value < 0.0001; ns—not significant.

**Figure 6 cells-10-01185-f006:**
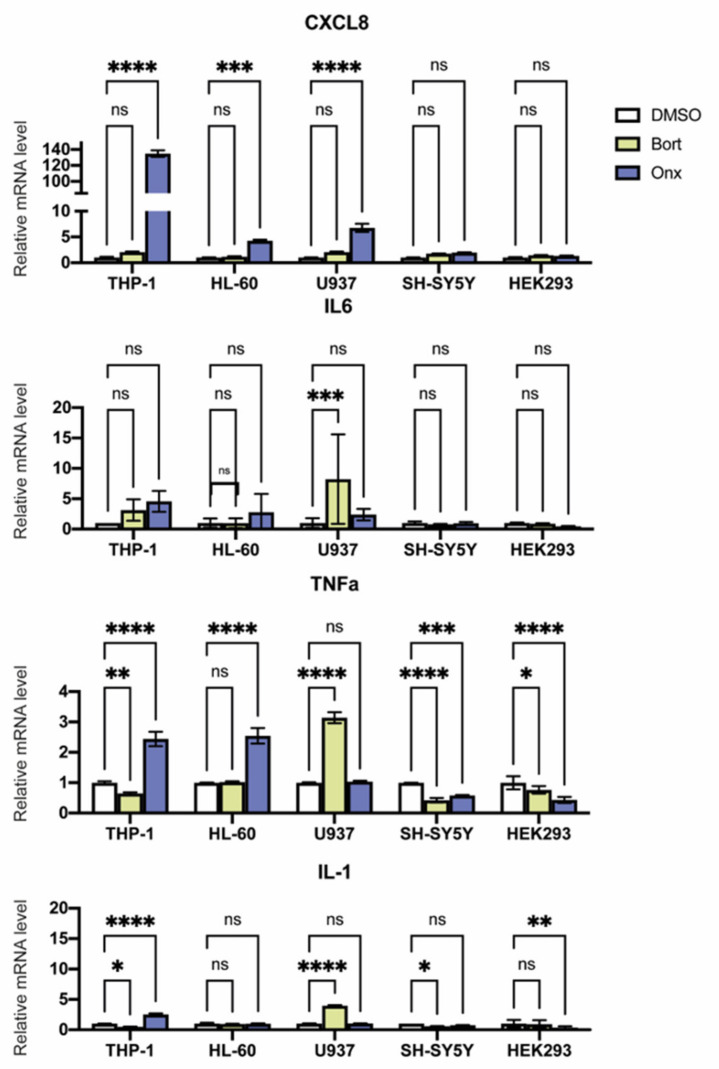
Relative, *IL-1β*, *IL-6*, *CXCL8*, and *TNF-α* mRNA levels in HL-60, U937, THP-1, HEK 293, and SH-SY5Y cells treated with Bortezomib or ONX-0914. Cells were treated with the inhibitors or DMSO for 24 h. After that, relative mRNA expression levels were determined by real-time PCR. Three technical repeats were performed for each sample. Two-way ANOVA test for multiple comparisons with no correction was used. Asterisks: * *p*-value < 0.1; ** *p*-value < 0.01; *** *p*-value < 0.001; **** *p*-value < 0.0001; ns—not significant.

## Data Availability

The data is available upon request.

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
