# Peer review of "Immunoproteasome Activity and Content Determine Hematopoietic Cell Sensitivity to ONX-0914 and to the Infection of Cells with Lentiviruses"

_cells, 2021, doi:10.3390/cells10051185_

Round 1

Reviewer 1 Report

Evaluation of the revised manuscript " Immunoproteasome Activity and Content Determines Hematopoietic Cell Sensitivity to ONX-0914 and to the Infection of Cells with Lentiviruses" by Elmira Vagapova et al.

The reviewer would like to thank the authors for their careful revision of the manuscript.

Several of my concerns were adequately addressed. There are, however, still some concerns remaining.

  1. The reviewer disagrees with the authors’ response regarding equipotent proteasome inhibitor doses. For sure this is a challenging task due to the different standard versus immunoproteasome population. However, the S-LLVY-AMC probe is a specific substrate for both the b5 and b51 subunit. As such it should be possible to adjust inhibitor concentrations to obtain a similar degree of CTL-site inhibition. If you would then choose equipotent doses e.g. for Thp1 cells there should be similar cell survival. Comparing 3 nM Bz with almost no cell death after 72h in Thp1 with 50 nM ONX with 50% cell death already suggests that there is a differential degree of CTL-inhibition as also suggested by Figure 3c. Accordingly, comparing viral transduction efficiency (Figure 3b) is based on a different extent of proteasome inhibition and thus not comparable in this regard.
  2. The authors show that the b5i substrate is only 3-fold more specific to the b5i compared to the b5 site (Figure 2 for the reviewer). This is not very specific. I would rather suggest to use b5-specific ABPs as for example published by the Overkleft lab (de Bruin et al., 2016). This allows a highly specific discrimination of the beta 5 active sites present in the cells and is better suited than the Ubiq ABP which did not fully resolve the b5 from the b5i and b1 versus b1i subunits (Figure 4 for the reviewer).
  3. Normalization of almost no b51 activity in HEK cells results in the overestimation of an effect that is not there as rightly pointed out by the authors. I would thus refrain from doing that. It suggests a result which is probably technically not robust ad almost not distinguishable from background signals. This is also suggested in GFigure 3c. ONX treatment of HEK cells reduces activity of the b5i site by 50%. As there is no b5i site present in HEK cells at baseline, this again is probably an unspecific effect which is overestimated by normalization.
  4. Lastly, reduction in transduction activity in Figure 5f is likely due to IFNg effects independent of the immunoproteasome as indicated by the massive baseline effects in DMSO controls. The authors should calculate the ratio of plus/minus IFNg for all conditions and determine whether the differences with the proteasome inhibitors are significant.

Author Response

Dear Reviewer 1, thank you so much for your comments and suggestions. The issues you have raised via performed new experiments lead us to interesting observations. Moreover, we would like to express appreciation for the Dr. Overkleeft paper which is especially useful for several of our research projects.

Please find our point-by-point responses below and see the attachment.

  1. The reviewer disagrees with the authors’ response regarding equipotent proteasome inhibitor doses. For sure this is a challenging task due to the different standard versus immunoproteasome population. However, the S-LLVY-AMC probe is a specific substrate for both the b5 and b51 subunit. As such it should be possible to adjust inhibitor concentrations to obtain a similar degree of CTL-site inhibition. If you would then choose equipotent doses e.g. for Thp1 cells there should be similar cell survival. Comparing 3 nM Bz with almost no cell death after 72h in Thp1 with 50 nM ONX with 50% cell death already suggests that there is a differential degree of CTL-inhibition as also suggested by Figure 3c. Accordingly, comparing viral transduction efficiency (Figure 3b) is based on a different extent of proteasome inhibition and thus not comparable in this regard.

According to the Reviewer suggestion we attempted to find equipotent concentrations of Bortezomib and ONX-0914. So, we used lower concentrations of the ONX-0914 (20 and 30 mM) and higher concentrations of Bortezomib (4 and 5 mM) and noticed several tricky things.

We have managed to obtain identical levels of transduction when 30 nM of ONX-0914 and 4 nM of Bortezomeb were used (Fig.1a). However, identical levels of transduction stimulation were obtained on a background of different degree of proteasome inhibition, thus 30 nM of ONX-0914 decreased chymotrypsin-like activity by 85% and β5i-specific activity by 90% while 4 nM of Bortezomib decreased proteasome activities by 70% (Fig. 2). Accordingly, mean fluorescence intensities were significantly higher in ONX-0914-treated cells (Fig.1b). Hence, we suggest that in the case of ONX-0914 inhibition of proteasomes likely also increases the multiplicity of infection while this parameter is not influenced by Bortezomib. Collectively this indicates that if we will have identical degree of proteasome inhibition, we might not have the identical levels of transduction. Indeed 20 mM of ONX-0914 decreased chymotrypsin-like activity by 80% and β5i-specific activity by 83% and 4 nM of Bortezomib decreased chymotrypsin-like activity by 70% (Fig. 2), but the transduction efficacy was higher in Bortezomib-treated cells (Fig.1a). Still the MFI was significantly higher in ONX-0914 treated cells than in Bortezomib treated cells (Fig. 1b). In our experiment we did not increase the concentration of Bortezomib further, since under these concentrations cells were already in a bad condition comparing to cells treated with the used concentrations of ONX-0914. Moreover, we detected a strange slight increase of proteasome activities in cells treated with 5 mM of Bortezomib comparing to cells treated with 4 mM of the same inhibitor. This might be associated with apoptosis; however, we did not verify that. Taken together our results indicated that when we find equipotent concentrations regarding the transduction, they will be not equipotent in the light of activity reduction. Another interesting conclusion that we can draw that in order to observe statistically significant transduction increase the proteasome activity should be decreased by more than 70%, indicating that the effect becomes obvious when more than 75-80% of initial proteasome activity is blocked.

Fig.1. Lentiviral transduction of THP-1 cells following treatment with different concentrations of ONX-0914 or Bortezomib. THP-1 cells were incubated with 20 and 30 nM of ONX-0914 or 4 and 5 nM of Bortezomib for 6 hours prior to the addition of lentiviral particles. After administration of viruses cells were incubated for additional 72 hours. Cells were collected and per cent of fluorescent cells (a) and mean fluorescence intensity (b) were measured using LSRFortessa flow cytometer (BD Biosciences). Dots represent individual replicates. Bars represent standard deviation. Three independent repeats were performed. Two-way ANOVA test for multiple comparisons with no correction was used. Asterisks: * - p-value <0.1; ** - p-value <0.01; *** - p-value < 0.001; **** - p-value <0,0001.

Fig. 2. Chymotrypsin-like and β5i-specific activities in THP-1 cells after incubation with proteasome inhibitors. Tests were performed in triplicates. Bars represent standard deviation.

In the main text we have noticed that: “It should be mentioned that since both ONX-0914 and Bortezomib affect immunoproteasomes, in cells with predominant immunoproteasome content at certain concentrations comparable effects may be expected. However, the normalization of the inhibitors concentrations is challenging due to several reasons including toxicity, specificity and targeting of constitutive proteasomes by Bortezomib. Nevertheless, by using lower concentrations of ONX-0914 and higher concentrations of Bortezomib we managed to obtain comparable transduction levels of THP-1 cells (Figure S4a, b), however the degree of proteasome inhibition (Figure S4c) and condition of cells were significantly different”. (Lanes 330-337 section 3.3. Results).

  1. The authors show that the b5i substrate is only 3-fold more specific to the b5i compared to the b5 site (Figure 2 for the reviewer). This is not very specific. I would rather suggest to use b5-specific ABPs as for example published by the Overkleft lab (de Bruin et al., 2016). This allows a highly specific discrimination of the beta 5 active sites present in the cells and is better suited than the Ubiq ABP which did not fully resolve the b5 from the b5i and b1 versus b1i subunits (Figure 4 for the reviewer).

Dear Reviewer, thank you so much for the suggestion and an exciting manuscript by de Bruin et al. I read it with immense interest. I included the paper in the reference list. In fact, we are very interested in having such probes for several our projects and I wrote a letter to Dr. Overkleeft if he perhaps can share them with us. We absolutely agree that Ubiq probe has several limitations, but unfortunately, the Ubiq-type probe was the only available to us for purchase. Another problem is that it takes at least 2-3 months to deliver such reagents in Russia.

  1. Normalization of almost no b51 activity in HEK cells results in the overestimation of an effect that is not there as rightly pointed out by the authors. I would thus refrain from doing that. It suggests a result which is probably technically not robust ad almost not distinguishable from background signals. This is also suggested in GFigure 3c. ONX treatment of HEK cells reduces activity of the b5i site by 50%. As there is no b5i site present in HEK cells at baseline, this again is probably an unspecific effect which is overestimated by normalization.

We have rearranged the Figures 3 and 5d and excluded signal normalization, thus reader would see that the actual levels of β5i activity in HEK cells are extremely low and statistically insignificant.

  1. Lastly, reduction in transduction activity in Figure 5f is likely due to IFNg effects independent of the immunoproteasome as indicated by the massive baseline effects in DMSO controls. The authors should calculate the ratio of plus/minus IFNg for all conditions and determine whether the differences with the proteasome inhibitors are significant.

We have rearranged Figure 5f and included the graph with normalization to the transduction levels, observed in DMSO-treated samples. We detected statistically significant increase of lentiviral transduction in IFN and ONX-0914 treated cells comparing to the levels revealed in cells after the ONX-0914 treatment only (Fig.3). 

Fig.3 Transduction efficacy of THP-1 cells pre-treated with proteasome inhibitors (either 50 nM of ONX-0914 or 3 nM of Bortezomib) and ether stimulated or not stimulated with the 1000 U/mL of IFN-γ for 48 hours (left panel). The same data were normalized to the level of transduction in DMSO-treated cells (right panel). Dots represent individual replicates. Bars represent standard deviation. Three independent repeats were performed. Two-way ANOVA test for mul-tiple comparisons with no correction was used. Asterisks: *-p-value <0.1; **-p-value <0.01; *** - p-value <0.001; **** - p-value <0.0001.

Reviewer 2 Report

The authors have addressed all my concerns and therefore,  I do not have any further comments. I feel that this paper is ready for publication.

Author Response

Dear Reviewer 2, thank you so much for reconsideration and high evaluation of our manuscript.

Round 2

Reviewer 1 Report

The authors have appropriately addressed all my concerns. I would like thank them for their replies. 

This manuscript is a resubmission of an earlier submission. The following is a list of the peer review reports and author responses from that submission.

Round 1

Reviewer 1 Report

Evaluation of the manuscript " Immunoproteasome Activity and Content Determines Hematopoietic Cell Sensitivity to ONX-0914 and to the Infection of Cells with Lentiviruses" by Elmira Vagapova et al.

In this manuscript, the authors investigate the contribution of the immunoproteasome on the infectivity of lentivirus in different hematopoietic and parenchymal cell lines of neuronal and kidney origin. For their analysis, they determine the RNA and protein expression levels of the different standard and immunoproteasomal catalytic subunits via RTqPCR and Western blotting, respectively. They also analyze the activity of the beta5 standard and beta5i immunoproteasome subunits using fluorescent substrates as well as a cell permeable activity based probe. The application of bortezomib (Bz) with broad specificity against standard and immunoproteasome catalytic subunits (mainly beta 5, 51, 1, 1i) compared to ONX-0914, which specifically targets the immunoproteasome subunits beta5i and beta1i, aims at the analysis of the distinct effects of immunoproteasome versus standard proteasome inhibition on virus replication and infectivity. From their data, the authors conclude that immunoproteasome activity might be an important determinant for antiviral resistance in cells with high immunoproteasome content. This concept, that the immunoproteasome is involved in limiting viral transduction efficiency, is interesting, novel and also very timely in light of the current COVID-19 pandemic. The conclusions, however, are not always supported by the data and alternative explanations need to be critically assessed.

Major concerns (the order does not imply any ranking):

  • The viability experiments need to take into account the degree of proteasome inhibition for the interpretation of the results. Proteasome activity needs to be assessed after 72 hours to interpret the susceptibility to proteasome inhibition. For example: the 3nM Bz does not inhibit the CTL-active sites in HEK cells to a large extent after 6 hours, while the 200nM ONX effectively inhibits the active sites in Thp1 cells. Accordingly, these doses of Bz and ONX-0914 are not equipotent on standard and immunoproteasome subunits, respectively. Applying them on the different cell lines and comparing their effect on virus transduction efficiencies is thus comparing apples and oranges.
  • The assessment of the inhibition of the different proteasome activities after 6 hours of treatment with BZ or ONX-0914 in Figure 4 is crucial for the interpretation of the data in Figure 2b. Accordingly, the data should be rearranged.
  • The data on expression of immunoproteasome subunits either at baseline or induced by INFgamma do not fit together with the activity levels as determined by the substrates for the chymotrypsin-like and β5i-specific proteasome activities. For example: In Figure 7b you nicely show the absence of beta51 and that it is also not induced by IFNgamma. You then show activity assays for this active site below. While Figure 4 indicates that the beta5i activity is much lower in HEK compared to Thp1 cells, there should be no specific activity at all. There are similar discrepancies with the other cell lines suggesting that the substrate is not specific for beta5i.
  • The dose of 1000U IFNgamma is very high and should be evaluated and compared to 100U.
  • The use of the cell permeable ABP that is used to measure the activity of the virus transduced cells needs to be validated first. This is not a very established method and needs to be controlled for the specific conditions applied.
  • Figure 4a: this does not fit with IP expression extent. In the hematopoietic cell lines there should be more beta5i activity. You would expect that also from the sensitivity analysis! This again indicates that the beta5i substrate is not specific.
  • Figure 4a Y axis: what is mkg?
  • check and change reference 2: this is not the right reference for the statement.

Reviewer 2 Report

In the manuscript, the authors demonstrate that inhibition of immunoproteasomes (iPs) in hematopoietic cells with ONX-0914 resulted in increased infection by lentivirus. They also revealed that activation of iPs by IFN-γ markedly reduced viral infectivity, which was rescued upon simultaneous iPs inhibition. Although the results presented here are potently interesting, the major problem with this paper is that several conclusions lack statistical support of any kind. The authors should carefully link their conclusions with the data presented. Also, I do have a few points that should be addressed by the authors. These are as follows.

Major points

  1. I am concerned about the lack of description of fluorescent-positive cells (Figure 2). What are the criteria for determining the fluorescence threshold from negative to positive during viral production? Because this result is a main finding of the study, I would encourage the authors to have viral titers alternatively quantified by the number of formed plaques.

  1. Many of the points made in the result section and conclusions are not convincing because they are not backed up sufficiently by the statistical analysis. Several graphs do not include any asterisks for significance (e.g. Figs 4A–C, 5A–D, 7D, 7F, 8). The authors should make a clear distinction between “significant” and “not significant” results. Accordingly, the result section and conclusions need to be revised robustly.

  1. Materials and methods: The author should apply an appropriate statistical test for comparison. A t-test is used to compare the means of two groups. When comparing three or more groups, t-test is not appropriate.

  1. There is not enough evidence to support that the immunoproteasome activity will determine the viral transduction. In order to define more clearly these relationships, the authors should show a correlative curve for immunoproteasome activity vs transduction efficacy in cell clones (Figure 5). Some would say the authors show only the ones that are convenient for the authors’ hypothesis.

  1. According to the information in the Introduction, ONX-0914 is a beta 5i specific inhibitor. In figure 3(b), PSMB8 (beta 5i) was expressed in SH-SY5Y cells but not in HEK293T cells. In figure 4(C), however, ONX-0914 decreased b5i-speficic activity in HEK293T cells but not in SH-SY5Y cells. This discrepancy needs to be explained.

  1. Related to figure 8, the paper could be strengthened by adding one or more cytokines.

  1. In my opinion, the results of figure 6 are unnecessary or redundant, because the increment of immunoprotesome activity was not confirmed in this study.

  1. In figure legend, it is not clear how many samples were measured in each experiment.

  1. English should be carefully revised.